# The Impact of Job Satisfaction on Creating a Sustainable Workplace: An Empirical Analysis of Organizational Commitment and Lifestyle Behavior

Ma. Janice J. Gumasing *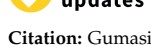 and Charles Kristian K. Ilo

School of Industrial Engineering and Engineering Management, Mapúa University, 658 Muralla St., Intramuros, Manila 1002, Philippines
* Correspondence: mjjgumasing@mapua.edu.ph; Tel.: +63-(2)-8247-5000

**Abstract:** The COVID-19 pandemic recently swept the globe, and quarantine sadly compelled most businesses and employees to adapt to the sudden change. Because of this, the employee may face psychological risks like a change in lifestyle, tiredness, burnout, and a drop in job satisfaction. Establishing how each aspect is associated with the job satisfaction of supply chain workers in the Philippines is the main topic of this study. Purposive sampling via an online survey is the non-probability sample method used in this investigation. The target respondents are the logistics company's employees. Google Forms were used to deliver the self-administered online survey questionnaire. The SEM model was assessed using the hypothesis test's beta coefficient and R2 findings. The model was proven sufficient to explain or forecast the employee's loyalty to the organization, stress at work, and job satisfaction. Results of the study revealed that organizational commitment (β = 0.716; *p*-value < 0.001) had the highest positive association with workers' job satisfaction, while work stress proved to have a negative association with job satisfaction (β = −0.166; *p*-value = 0.039). Similarly, job characteristics (β = 0.684; *p*-value < 0.001) and job involvement (β = 0.189; *p*-value = 0.009) were also proved to have a strong positive association with organizational commitment, which in turn influences job satisfaction. Finally, lifestyle behavior was found to have a positive association with work stress (β = 0.467; *p*-value < 0.001) and job burnout (β = 0.369; *p*-value = 0.001), negatively influencing job satisfaction. As a result, this study can offer supply-chain company personnel more information on the effect of organization commitment, lifestyle behavior, work stress, and job burnout on job satisfaction, which is a key component of job sustainability. Companies must ensure that employees are treated well and meet their demands to promote a sustainable workplace.

**Keywords:** burnout; job satisfaction; lifestyle behavior; organizational commitment; work stress; PLS-SEM

## 1. Introduction

The coronavirus disease (COVID-19) has spread worldwide since its first incidence in Wuhan, China, in December 2019. The COVID-19-induced global economic downturn has increased job insecurity among the workforce, and temporary work arrangements and unpaid leave have proliferated [1]. COVID-19 has forced numerous workers to adjust to new norms, behaviors, commute patterns, preferences, and work-from-home requirements. A sizable portion of employees continue to work from home due to the many abrupt changes to daily schedules and commutes that occurred, primarily in the early months of the COVID-19 pandemic, because employers and organizations were constantly promoting this pattern to stop the spread of infections and ensure the health and safety of workers [2]. Due to these restrictions, the number of employees is restricted from working on-site at any time, and many organizations have applied flexible work arrangements during this pandemic [3].

During the COVID-19 pandemic, many organizations implemented flexible work arrangements to adapt to the challenges posed by the crisis. This included logistic workers who played a big role in ensuring the smooth flow of essential goods and services. In the Philippines, logistic workers were allowed to work remotely, especially for administrative tasks, data analysis, or coordination roles. Remote work enabled employees to carry out their responsibilities from home, reducing the risk of exposure to the virus while maintaining productivity.

Shift rotation was also implemented to minimize the number of workers present at the workplace at any given time. The workforce was divided into smaller groups so that employees could work alternate shifts or days, ensuring continuous operations while adhering to social distancing guidelines. Additionally, given the high demand for logistic workers during and after the pandemic, companies implemented staggered work hours to allow logistic workers to arrive and depart at different times, reducing congestion during peak hours and minimizing contact between employees.

The COVID-19 pandemic has brought on unprecedented levels of business disruption, and sharp changes in supply and demand structures have affected all industries. The pandemic disrupted markets so quickly that businesses could not create and implement risk-aversion strategies, leaving them unprepared for the disruption [4]. While supply and demand are disrupted, companies—and specifically, the employees under them—may experience lifestyle changes, exhaustion, burnout, and a different level of job satisfaction compared to before the pandemic.

The pandemic quickly took over the world and altered all facets of life, including social interactions, working habits, commercial practices, and even lifestyles. The pandemic increased the inherent psychological unpredictability that comes with life and is linked to the virus's contagiousness [5]. Organizations and individuals have become aware of the detrimental effects of job dissatisfaction, occupational stress, and unhealthy lifestyle choices, which can impact productivity, healthcare quality, well-being, and workers' psychosocial risks [6].

Psychosocial risks can impact psychological and physical health through stress-activated psychophysiological pathways [7]. This unfavorable psychological condition in workers can be brought on by excessive demands and emotional requirements, and by a lack of personal or material resources to deal with them. Job burnout and work-related stress are examples of these. According to Salvagioni [8], burnout is a syndrome that results from chronic stress at work, with several consequences for workers' well-being and health. Burnout is typically experienced by individuals who have been exposed to prolonged periods of high stress, excessive workload, or challenging work environments. Job burnout is characterized by feelings of emotional exhaustion, cynicism, and a lack of effectiveness at work. According to Maslach et al. [9], burnout has been linked to several adverse affective outcomes, such as a decline in job satisfaction, which can negatively impact work performance, loyalty to the organization, and employees' desire to remain in the organization. Although job burnout among workers has been extensively researched, only some of these studies look at how burnout during the COVID-19 pandemic affects job satisfaction.

Job satisfaction is a defense against risks that predict generalized labor stress. It is essential to underline that effective management of psychosocial risks appears to be linked to it. The degree to which people enjoy and find fulfillment in their professions can be characterized as job satisfaction [10]—or the amount to which an employee feels that their employer is meeting their requirements [11]. Various theoretical models may describe the aspects that influence high job satisfaction.

Job satisfaction is also influenced by one of the most severe risks to employees' workplace health and work stress. Work stress refers to the physical, mental, and emotional strain experienced by individuals in response to work-related demands and pressures [12]. Work stress can arise from various factors, including the nature of the job itself, work environment, organizational culture, interpersonal relationships, and individual character-

istics [13]. Because of the pitiful mismatch between the workforce's capacity, job demands, and requirements, occupational stress is the inability to handle a challenge or pressure that arises [14]. A study indicates that work stress influences unsafe behaviors favorably, both directly and indirectly, through fatigue. Additionally, it was found that a safe workplace climate reduced the link between risky behaviors and work-related stress [15].

Previous research has found a strong relationship between workplace stress and job satisfaction. Low levels of job satisfaction are correlated with high levels of work stress. Job stress is associated with job discontent and a higher likelihood of leaving the company. An employee's affective orientation toward their work is reflected in their level of job satisfaction. There is considerable evidence that current employment patterns may negatively impact job satisfaction and worsen workers' physical and mental health [16].

It was discovered that work-related stress is a problem of growing concern in developing countries and that numerous studies on the topic have been conducted. Stress is the second most frequently reported work-related health issue in Europe; it accounts for 50–60% of all missed workdays, and the number of people with stress-related illnesses brought on by or made worse by their jobs is certain to rise [17]. This has a high cost in terms of increased human suffering and decreased economic performance, especially during COVID-19 [18].

COVID-19 also interfered with establishing and sustaining workers' healthy lifestyle practices, such as regular physical activity, getting enough sleep, and socializing with friends and family. Much has been written about the COVID-19 pandemic and decreased physical activity levels [19]. Engaging in physical activity can significantly impact one's health, including weight loss, better sleep, a strengthened immune system, and the ability to manage stress [20]. Reduced sleep length and poor sleep quality are linked to increased levels of perceived stress and are detrimental to the quality of life [21]. Manufacturing workers, particularly logistics workers, are thought to be at significant risk of having a low quality of life.

Job satisfaction is one of the many individual mechanisms that directly influence work performance and organizational commitment. Organizational commitment refers to an individual's psychological attachment, loyalty, and identification with their organization [21]. It reflects the extent to which employees feel a sense of belonging, involvement, and dedication towards their organization's goals and values. According to Dinc [22], organizational commitment is an important factor that influences employee attitudes, behaviors, and overall job performance. Employees may perform their jobs better and decide to stay with the company for a longer period if they are extremely satisfied with their jobs and feel happy while working. Organizational commitment and job satisfaction are strongly related. Additionally, it has a significant positive correlation with life satisfaction [23].

In shaping the organizational commitment, job characteristic and job involvement may play a significant role. Job characteristics refer to the aspects of a job that can impact employee experiences and motivation [24]. According to the Isfahani et al. [25], certain job characteristics, when present, can enhance employee satisfaction and motivation. Research has shown that higher levels of job characteristics, such as skill variety, task identity, task significance, autonomy, and feedback, are positively associated with organizational commitment [26]. Job involvement, on the other hand, refers to the psychological identification and engagement individuals have with their work [27]. It reflects the extent to which employees are emotionally connected to and invested in their job. According to Jose and Mampilly [28], employees with high job involvement demonstrate a strong sense of interest, enthusiasm, and dedication towards their work. Job involvement has a positive relationship with organizational commitment. When employees are highly involved in their job, they are more likely to develop a strong commitment to the organization [29]. Thus, understanding and addressing these factors can help organizations foster a more committed and engaged workforce.

Job dissatisfaction, workplace stress, and unhealthy lifestyle choices among employees have a negative impact on both the organization and the individual level. All things

considered, there is a pressing need for increased focus on workers' job satisfaction that will aid workers through this challenging time caused by the COVID-19 pandemic. However, studies have yet to be conducted to examine the factors influencing workers' job satisfaction directly. Studies about a collaborative relationship between job satisfaction, job burnout, work stress, lifestyle behavior, and organizational commitment have limited sources.

Given these significant changes in the work setting after the pandemic, we intend to prove in our study the effect of lifestyle behavior—which according to studies, significantly changed after the pandemic—and organizational commitment, given the changes in the company work setting and environment on job satisfaction among logistic workers. Thus, the main objective of this study is to fill the gap in the literature regarding supply chain employees and their status with psychosocial risks and some of its main consequences, such as job burnout and work stress, as well as work organization and lifestyle, on job satisfaction after the COVID-19 pandemic. Furthermore, the data gathered and tested in this study will determine which underlying factor contains the most influence on job satisfaction. Using the findings from this study, a few recommendations will be made in order to contribute to the level of job satisfaction of employees.

## 2. Materials and Methods

### 2.1. Conceptual Framework

Job satisfaction is one of the essential components of work motivation, which is a significant factor in determining organizational behavior. The literature and empirical studies on job satisfaction are extensive [30]. There have been numerous aspects that have been separately discussed that will affect job satisfaction. Limitations on personal and professional development, work features, employment stability, organizational support, interpersonal relationships inside the organization, ties to one's immediate superior, and other considerations are among them [31]. Numerous factors are either positively or adversely related to job satisfaction. The four most common factors—work stress, job burnout, organizational commitment, and lifestyle behavior—have been the subject of this study. Stress at work is a critical component of job satisfaction. When stress at work serves as a motivator, creativity and fulfillment are produced, eliminating boredom and monotony. When stress plays a negative role in job burnout, it causes hostility and low job satisfaction [32]. Job burnout is a form of personal stress syndrome contextualized within intricate social connections at work and is likely to result in a lack of job satisfaction. Earlier work also noted that lifestyle choices and burnout symptoms are related [33]. Physical symptoms can be brought on by insufficient rest, a bad diet, and a lack of leisure for workers [34].

On the other hand, there has been a fair amount of research on the connection between organizational commitment and work satisfaction. Although there is agreement among researchers that there is a connection, there is disagreement over the nature of the relationship [35]. Organizational commitment measures how much individuals identify with the company where they work, how involved they are, and whether they are prepared to leave the company [36]. According to studies, job satisfaction predicts organizational commitment [37–39] and organizational commitment is a prerequisite for job satisfaction [40]. This study developed a theoretical framework by considering the chosen factors, such as work stress, job burnout, organizational commitment, and lifestyle behavior, to determine their impact on job satisfaction. The conceptual framework is shown in Figure 1.

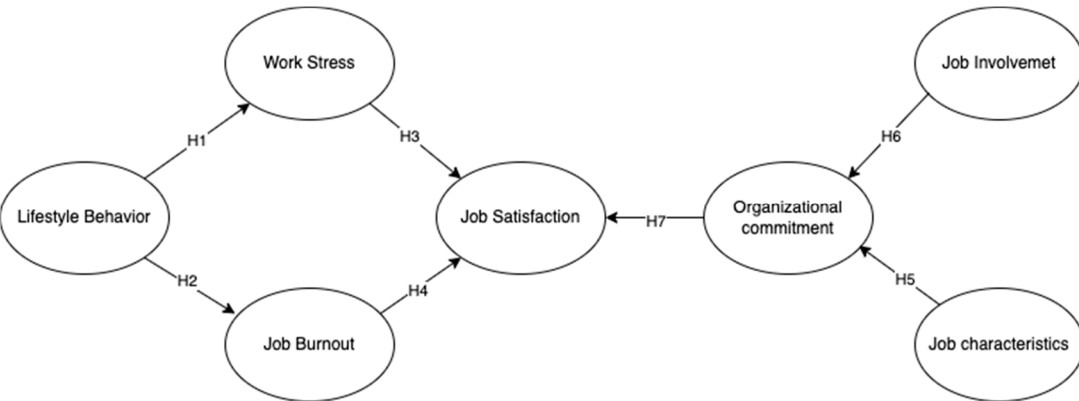

**Figure 1.** Conceptual Framework to Determine Factors Affecting Job Satisfaction.

*2.2. Determinants of Job Satisfaction*

Lifestyle behaviors are routine actions that come from a person's values, knowledge, and norms that are influenced by a larger cultural and social context. These behaviors, which are impacted by a range of social traits, influence body weight and general health [41]. Lifestyle behaviors, including smoking, poor diet, physical inactivity, and drinking too much alcohol, are all considered factors of health risks [42]. These behaviors concern employees and may be connected to psychological problems, including job burnout and stress [43]. Prior studies have proved the significant association between poor lifestyle, work stress, and job burnout. Rawat's [44] study showed increased weight gain, sugar levels and stress levels caused by poor lifestyle behavior. Harma [45] revealed that a poor lifestyle, such as having insufficient or poor sleep, is related to work stress, which may negatively impact job satisfaction. In a study by Shubayr et al. [6], a negative and moderate relationship was found between job satisfaction and unhealthy lifestyle behavior. Moreover, burnout relationships with job satisfaction and healthy lifestyle behaviors were also found to have negative and moderate relationships. Thus, it was hypothesized that:

**H1.** *Poor lifestyle behavior has a significant and positive effect on work stress.*

**H2.** *Poor lifestyle behavior has a significant and positive effect on job burnout.*

**H3.** *Work stress has a significant and negative effect on job satisfaction.*

**H4.** *Job burnout has a significant and negative effect on job satisfaction.*

Organizational commitment reflects the extent to which individuals identify with the company where they work, their level of engagement, and if they are prepared to leave the company [46]. Studies suggest that job involvement and characteristics are the antecedents to organizational commitment [35]. In the present study, job involvement represents the attitude toward work typically characterized as the degree of psychological identification with one's work. In contrast, job characteristics are based on the notion that a task is the source of an employee's motivation. In other words, a tough, diverse job increases motivation, while a monotonous, repetitive job causes stress detrimental to the worker's job satisfaction. Numerous studies have shown a significant link between organizational commitment and job satisfaction [47–50]. Thus, it was hypothesized that:

**H5.** *Job involvement has a significant and positive effect on organizational commitment.*

**H6.** *Job characteristic has a significant and positive effect on organizational commitment.*

**H7.** *Organizational commitment has a significant and positive effect on job satisfaction.*

### 2.3. Methodology

This study used a non-probability sampling technique, specifically purposive sampling using an online survey. The employees of the logistics company are the target respondents. The self-administered questionnaires for the online survey were provided via Google Forms. The survey link was given to the target respondents for 3 months using various social media platforms, including Instagram, Facebook, Twitter, WeChat, and Viber. The questionnaires were distributed using a variety of cross-sectional designs. The total number of respondents is 400, following the study of German et al. [51], where the level margin of error is set at 10%. A total of 500 survey questionnaires were distributed to the target participants. However, only 400 participants responded to the study, resulting in a response rate of 80%.

The survey consists of a 44-item question. It was presented as one questionnaire having two parts. The first part of the questionnaire determines the respondent's demographic profile, including age, gender, educational attainment, area of residence, work position, employment status, and monthly salary/income.

The second part of the questionnaire consists of the indicators based on a framework for job satisfaction. It consists of item questionnaire where all answers were on a 5-point Likert scale ranging from strongly disagree to strongly agree. Seven latent were used in the study, including lifestyle behavior, work stress, job burnout, job involvement, job characteristics, organizational commitment, and job satisfaction. The measures for each latent were developed and adapted based on previous studies. The construct and measurement items are shown in Table 1.

**Table 1.** The construct and measurement items.

| Construct | Items | Measure | Supporting Reference |
|---|---|---|---|
| Lifestyle Behavior | LB1 | I spend most days being physically inactive | [52,53] |
| | LB2 | I am concerned about my current weight | |
| | LB3 | I have insufficient and poor sleep quality | |
| | LB4 | I frequently drink beverages with high content of sugar | |
| | LB5 | I regularly smoke tobacco or cigarette | |
| | LB6 | I frequently drink alcoholic beverages | |
| Work Stress | WS1 | I work with two or more groups that operate quite differently | [54–56] |
| | WS2 | I receive conflicting tasks from two or more people | |
| | WS3 | I work on unnecessary things | |
| | WS4 | I am not aware of my responsibilities | |
| | WS5 | It Is not clear to me what must be done | |
| | WS6 | I do not exactly know what is expected of me | |
| Job burnout | JB1 | I feel emotionally drained from my work | [57–59] |
| | JB2 | I feel exhausted by the end of the workday | |
| | JB3 | I feel overtired when I wake up and must experience another day in the job | |
| | JB4 | I feel burned out due to my work | |
| | JB5 | I feel frustrated by my job | |
| Job involvement | JI1 | I am very much involved personally in my job | [60–62] |
| | JI2 | I live, eat and breath my job | |
| | JI3 | Most of my interests are centered around my job | |
| | JI4 | Most of my personal life goals are job-oriented | |
| | JI5 | I consider my job to be very central to my existence | |

**Table 1.** *Cont.*

| Construct | Items | Measure | Supporting Reference |
|---|---|---|---|
| Job Characteristics | JC1 | I have stimulating and challenging work | [63–65] |
| | JC2 | I have great chances to exercise independent thought and action | |
| | JC3 | I have opportunities to learn new things in my work | |
| | JC4 | I have opportunities for personal growth and development | |
| | JC5 | I have a sense of worthwhile accomplishment in my work | |
| Organizational Commitment | OC1 | I talk up this organization to my friends as a great organization to work for | [35,66,67] |
| | OC2 | I am proud to tell others that I am parts of this organization | |
| | OC3 | I am extremely glad that I chose this organization to work | |
| | OC4 | For me this is the best of all possible organizations for which to work | |
| Job Satisfaction | JS1 | The company clearly conveys its mission to me | [35,67,68] |
| | JS2 | I have the tools and resources I need to do my job | |
| | JS3 | The amount of work expected of me is reasonable | |
| | JS4 | It is easy to get along with my colleagues | |
| | JS5 | The morale in my department is high | |
| | JS6 | Overall, I feel satisfied with my job | |

To validate the question items, a pilot test was conducted to assess the questionnaire's validity and reliability using SPSS 24 prior to distributing the survey. Initial survey responses of 190 samples were used in the pilot test who were not considered for the main study. The sampling size met the recommended sample size of 5 subjects per item, or at least 185 participants for a 37-item questionnaire. They were asked to answer the questionnaire, and the researcher was always available to clarify any phrases or terms.

After gathering the initial data, an exploratory factor analysis (EFA) and measure of internal consistency using Cronbach's alpha were used to validate the items in the questionnaire. Data suitability for EFA were met by having a Kaiser–Meyer–Olkin (KMO) value of $>0.7$ and a significant $p$-value $< 0.05$ for Bartlett's test of sphericity for all constructs in the questionnaire. The number of factors were determined using an eigenvalue greater than 1.0 and a visual inspection of the scree plot. The factors were extracted using the principal axis factoring method. Cronbach's alpha coefficient was used. The Cronbach's alpha value was found to be more than 0.70. Furthermore, Harman's Single Factor Test was also used to examine if there was any common method bias (CMB). With a value of 23.41%, the test showed no CMB. Afterward, the questionnaire was disseminated to obtain the final data.

### 2.4. Structural Equation Modeling

The study used a causal inference theory focusing on establishing a cause-and-effect relationship between variables. This theory provides a systematic approach to test and understand the direct relationship between independent and dependent variables. They help researchers determine whether changes in the independent variable have a significant impact on the dependent variable, allowing for a deeper understanding of the underlying mechanisms and effects. Various statistical techniques such as structural equation modeling can be used to test and establish causal relationships.

Multivariate analysis was used to examine the survey data, explicitly utilizing Smart PLS version 3.3.3. To determine the association between the various variables and the level of job satisfaction among the workers, partial least square structural equation modeling (PLS-SEM) was used on the data gathered. PLS-SEM is increasingly used in scientific research and studies because it varies from other modeling approaches in that it investigates the direct and indirect impacts on presumed causal links [69].

PLS-SEM is a variance-based modeling and multivariate analytic tool that is widely used to simultaneously link several indicators or constructs, according to Ullman [70]. As an alternative to variance-based SEM utilizing AMOS, PLS-SEM is used to identify essential indicators and constructs and investigate the link of an existing structural theory [71]. Standardized root mean square residual (SRMR), normal fit index (NFI), and chi-square were also used to demonstrate the model fit in this study utilizing PLS-SEM. A value of less than 0.08 is considered a good fit for SRMS [72]. MacCallum [73] state that, for NSI, a score of 0.90 and above denotes an acceptable fit, while for chi-square, a number lower than 5.0 denotes a model that fits the data well [74].

The significance level of the path coefficients and the $R^2$ measurements was established in the study's model using an $R^2$ value of 0.20 and above. An $R^2$ value of 0.20 is regarded as high, according to Hair et al. [71]. In order to quantify the relationship between various factors, path analysis was also done to determine the causal relationship among the variables [75]. Path analysis' main goal is mediation, which argues that a variable can directly and indirectly affect a result through another variable [69].

## 3. Results

### 3.1. Demographic Profile

Table 2 summarizes the demographic profile of the respondents based on gender, age, educational attainment, area of residence, work position, employment status and monthly salary/income. It is observed that 54% of the respondents are male and 46% are females. The majority of those who responded are 31 to 40 years old, representing 38% of the respondents, followed by 41- to 50-year-olds and 21- to 30-year-olds, with a percentage of 28% and 26%, respectively, while only 7.5% are near seniority—that is, ages 51 and above. Only 0.5% of the respondents are aged 20 and below. In total, 69.5% of the respondents are college graduates, 17% are only high school graduates, 13% have finished graduate studies, and only 0.5% were able to finish elementary. Most participants are from cities with a percentage of 63%, while 37% are from provinces. For work positions, most participants are directly working as logistics personnel (54%), while 15.5% are in order management, followed by customer service with a percentage of 12%. In terms of work status, only 1.5% of the respondents are part-time workers while the majority are full-time employees with a percentage of 98.5%. Lastly, these workers' monthly income or salary averaged Php 15,000 to Php 29,999 having 27.5% of the participants, followed by Php 30,000 to Php 59,999 income and Php 14,999 or less income, with percentages of 22.5% and 22%, respectively. In comparison, only 2% have an income higher than Php 200,000.

**Table 2.** Summary Statistics of Demographic Profile.

| Respondent's Profile | Category | N | % |
|---|---|---|---|
| Gender | Male | 215 | 54% |
| | Female | 185 | 46% |
| Age | 20 and Below | 2 | 0.5% |
| | 21–30 | 204 | 26.0% |
| | 31–40 | 152 | 38.0% |
| | 41–50 | 112 | 28.0% |
| | 51 and above | 30 | 7.5% |
| Educational Attainment | Elementary Graduate | 2 | 0.5% |
| | Highschool Graduate | 68 | 17.0% |
| | College Graduate | 278 | 69.5% |
| | Others | 52 | 13.0% |
| Area of Residence | City | 252 | 63% |
| | Province | 148 | 37% |

**Table 2.** *Cont.*

| Respondent's Profile | Category | N | % |
|---|---|---|---|
| Work Position | Order Management | 62 | 15.5% |
| | Customer Service | 48 | 12.0% |
| | Warehouse Manager | 30 | 7.5% |
| | Transport Manager | 12 | 3.0% |
| | Inventory Manager | 0 | 0% |
| | Billing Clerk | 4 | 1.0% |
| | Receiving Supervisor | 12 | 3.0% |
| | Load Planner | 8 | 2.0% |
| | Dispatcher | 8 | 2.0% |
| | Others | 216 | 54.0% |
| Employment Status | Full Time | 394 | 98.5% |
| | Part Time | 6 | 1.5% |
| Monthly Salary/Income | Php 14,999 or Less | 88 | 22.0% |
| | Php 15,000 to Php 29,999 | 110 | 27.5% |
| | Php 30,000 to Php 59,999 | 90 | 22.5% |
| | Php 60,000 to Php 99,999 | 58 | 14.5% |
| | Php 100,000 to Php 199,999 | 46 | 11.5% |
| | Php 200,000 or more | 8 | 2.0% |

*3.2. Result of SEM*

Figure 2 illustrates the initial SEM for the employee's job satisfaction and several factors stated by several studies. This working model, as constructed, demonstrates the relationships of factors in connection to job satisfaction. The questionnaires serve as the measuring factor for each outer variable connected to job satisfaction. The questionnaire contains several items, and Figure 2 shows the observed value for each factor. This conceptual model guides the analysis of this study using the structured equation modeling. The subsequent tests determine whether the hypothesis is true and accurate.

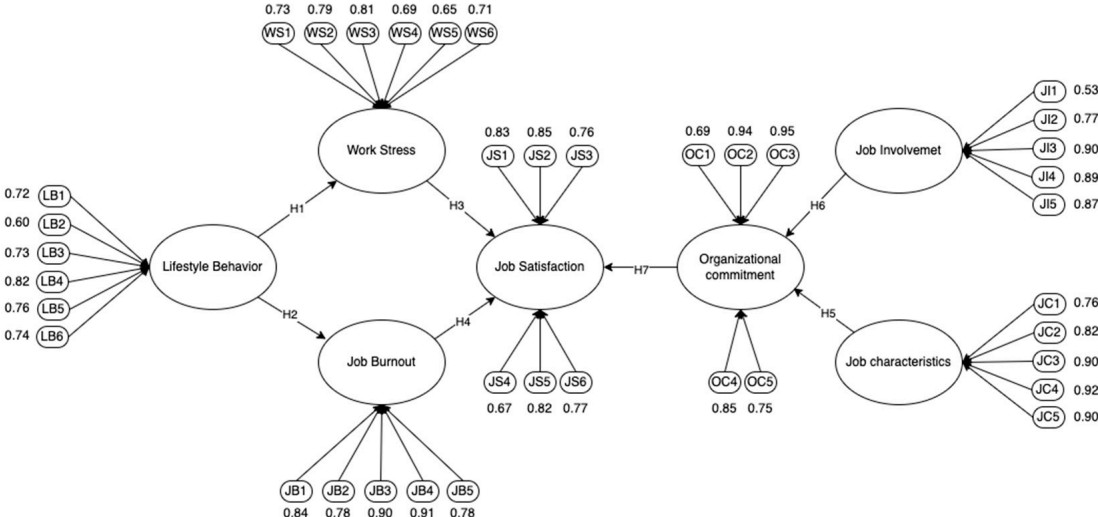

**Figure 2.** Initial SEM for Determining the Factors Affecting Job Satisfaction.

Table 3 displays the values for validity and reliability checking of the observed data through the running model. Cronbach's alpha ($\alpha$) tests for the consistency of each item of the variables determines their significance in the working model. Apart from Cronbach's, composite reliability (CR) and average variance extracted (AVE) were also used to test the variables' reliability and validity, respectively. The grand mean value of the squared loadings of the indicators connected to the underlying construct is used to calculate the average variance extracted. Meanwhile, convergent validity investigates the construct's

validity to the extent that it converges to account for the variation in its indicators. As a result, several items had not sufficed to capture the dormant variable's variability. Items that contained initial loading values less than 0.7 were not considered in the testing. The AVE's threshold value for convergent validity should be greater than 0.5 [71]. All values are higher than required, which leads to increased internal consistency and dependability across the test items sample. This suggests that all the model's constructs are valid and dependable [51].

**Table 3.** Reliability and convergent validity result.

| Construct | Items | Mean | S.D. | FL ($\geq$0.7) | $\alpha$ ($\geq$0.7) | CR ($\geq$0.7) | AVE ($\geq$0.5) |
|---|---|---|---|---|---|---|---|
| Lifestyle Behavior | LB1 | 3.62 | 1.15 | 0.723 | 0.783 | 0.855 | 0.718 |
| | LB2 | 3.69 | 1.17 | 0.600 | | | |
| | LB3 | 3.33 | 1.03 | 0.730 | | | |
| | LB4 | 2.57 | 1.22 | 0.818 | | | |
| | LB5 | 1.46 | 1.07 | 0.763 | | | |
| | LB6 | 1.92 | 1.04 | 0.765 | | | |
| Work Stress | WS1 | 3.43 | 1.29 | 0.731 | 0.732 | 0.893 | 0.768 |
| | WS2 | 2.84 | 1.19 | 0.792 | | | |
| | WS3 | 2.19 | 1.06 | 0.813 | | | |
| | WS4 | 1.54 | 1.05 | 0.692 | | | |
| | WS5 | 1.42 | 0.78 | 0.653 | | | |
| | WS6 | 1.70 | 1.01 | 0.706 | | | |
| Job Burnout | JB1 | 2.43 | 1.05 | 0.842 | 0.898 | 0.908 | 0.713 |
| | JB2 | 2.67 | 1.16 | 0.781 | | | |
| | JB3 | 2.39 | 1.06 | 0.895 | | | |
| | JB4 | 2.27 | 1.03 | 0.913 | | | |
| | JB5 | 1.92 | 0.98 | 0.780 | | | |
| Job Involvement | JI1 | 4.17 | 0.90 | 0.534 | 0.853 | 0.884 | 0.643 |
| | JI2 | 3.45 | 1.09 | 0.765 | | | |
| | JI3 | 3.04 | 1.11 | 0.898 | | | |
| | JI4 | 3.05 | 1.11 | 0.885 | | | |
| | JI5 | 3.03 | 1.18 | 0.869 | | | |
| Job Characteristics | JC1 | 3.95 | 0.88 | 0.764 | 0.914 | 0.932 | 0.745 |
| | JC2 | 3.88 | 0.90 | 0.817 | | | |
| | JC3 | 4.24 | 0.87 | 0.903 | | | |
| | JC4 | 4.22 | 0.86 | 0.924 | | | |
| | JC5 | 4.18 | 0.82 | 0.897 | | | |
| Organizational Commitment | OC1 | 3.78 | 0.93 | 0.692 | 0.881 | 0.906 | 0.746 |
| | OC2 | 4.26 | 0.91 | 0.939 | | | |
| | OC3 | 4.20 | 0.89 | 0.951 | | | |
| | OC4 | 4.05 | 0.89 | 0.847 | | | |
| | OC5 | 3.98 | 0.92 | 0.752 | | | |
| Job satisfaction | JS1 | 3.96 | 0.90 | 0.833 | 0.874 | 0.883 | 0.615 |
| | JS2 | 3.96 | 0.97 | 0.845 | | | |
| | JS3 | 3.69 | 0.95 | 0.760 | | | |
| | JS4 | 4.05 | 0.89 | 0.666 | | | |
| | JS5 | 3.95 | 0.95 | 0.819 | | | |
| | JS6 | 4.14 | 0.79 | 0.770 | | | |

Examining discriminant validity has become necessary before examining correlations between latent variables. The Fornell–Larcker criterion and cross-loading investigation are the prevalent methods for assessing discriminant validity in variance-based structural equation modeling, such as partial least squares [76]. Discriminant validity is proven when, using variance-based SEM for the Heterotrait–Monotrait ratio, a value between two reflective constructs falls below 0.85, and the assigned constructs have a larger value than all loadings of other constructs for Fornell–Larcker [77]. According to Tables 4 and 5, the

findings demonstrate satisfactory reliability and convergent validity, and the values fall within the intended range. Thus, the overall results across the constructs are accepted. The conventional metric of comparing the squared AVE of each latent variable to all other reflectively measured latent variables in the structural model was first proposed by Fornell and Larcker [78]. All model constructs' shared variances should not have greater value than their squared AVEs. Table 4 demonstrates that almost all latent variables have higher squared AVEs than the correlation coefficients of other latent variables. This shows that the model has a good level of convergent, reliable, and discriminant validity.

**Table 4.** Discriminant Validity: Fornell–Larcker Criterion.

|      | JB    | JC    | JI    | JS    | LB    | OC    | WS    |
|------|-------|-------|-------|-------|-------|-------|-------|
| JB   | 0.813 |       |       |       |       |       |       |
| JC   | 0.650 | 0.770 |       |       |       |       |       |
| JI   | 0.626 | 0.560 | 0.798 |       |       |       |       |
| JS   | 0.622 | 0.648 | 0.638 | 0.766 |       |       |       |
| LB   | 0.592 | 0.674 | 0.594 | 0.675 | 0.810 |       |       |
| OC   | 0.663 | 0.713 | 0.560 | 0.623 | 0.591 | 0.827 |       |
| WS   | 0.581 | 0.677 | 0.634 | 0.712 | 0.781 | 0.604 | 0.821 |

**Table 5.** Discriminant Validity: Heterotrait–Monotrait Ratio.

|      | JB    | JC    | JI    | JS    | LB    | OC    | WS    |
|------|-------|-------|-------|-------|-------|-------|-------|
| JB   |       |       |       |       |       |       |       |
| JC   | 0.781 |       |       |       |       |       |       |
| JI   | 0.790 | 0.780 |       |       |       |       |       |
| JS   | 0.827 | 0.709 | 0.798 |       |       |       |       |
| LB   | 0.790 | 0.659 | 0.728 | 0.671 |       |       |       |
| OC   | 0.648 | 0.747 | 0.834 | 0.802 | 0.683 |       |       |
| WS   | 0.562 | 0.810 | 0.808 | 0.673 | 0.646 | 0.709 |       |

Two latent variables that represent various theoretical notions are statistically different when they have discriminant validity. The Fornell–Larcker criterion is a regularly used method for evaluating discriminant validity [78]. Henseler, Ringle, and Sarstedt [76] developed a novel method for evaluating discriminant validity: the heterotrait–Monotrait ratio of correlations (HTMT), which gauges how comparable latent variables are. Discriminant validity can be established if the HTMT is less than one. A threshold of 0.85 reliably separates those pairings of latent variables that are discriminant and valid from those not in numerous real-world scenarios. Monte Carlo simulations show that the HTMT performs well in classification [79,80]. As observed in Table 3, only occupational characteristics concerning job involvement with a near value of 0.834 to 0.85, which is close to a lack of discriminant validity.

The method performed was PLS-SEM, a frequently used variance-based modeling and multivariate analytical method to simultaneously link various variables or constructs, according to Ullman [70]. As shown in Table 6, with a beta coefficient 0.716, organizational commitment has the highest significant influence on job satisfaction while wok stress has significant negative effect on job satisfaction having beta coefficient of −0.166. Similarly, job characteristics (β = 0.684; *p*-value < 0.001) and job involvement (β = 0.189; *p*-value = 0.009) were also proved to have a significant positive effect on organizational commitment, which in turn influences job satisfaction. Finally, lifestyle behavior was found to have a positive association with work stress (β = 0.467; *p*-value < 0.001) and job burnout (β = 0.369; *p*-value = 0.001). In contrast, the relationship between job burnout and job satisfaction at a *p*-value of 0.075 had shown to be not significant, and therefore, resulting in a rejection of the hypothesis.

**Table 6.** Hypothesis Test.

| No | Relationship | Beta Coefficient | *p*-Value | Result | Significance | Hypothesis |
|---|---|---|---|---|---|---|
| 1 | LB → WS | 0.467 | <0.001 | Positive | Significant | Do not Reject |
| 2 | LB → JB | 0.369 | 0.001 | Positive | Significant | Do not Reject |
| 3 | WS → JS | −0.166 | 0.039 | Negative | Significant | Do not Reject |
| 4 | JB → JS | −0.139 | 0.075 | Negative | Not Significant | Reject |
| 5 | JI → OC | 0.189 | 0.009 | Positive | Significant | Do not Reject |
| 6 | JC → OC | 0.684 | <0.001 | Positive | Significant | Do not Reject |
| 7 | OC → JS | 0.716 | <0.001 | Positive | Significant | Do not Reject |

*3.3. Model Fit Analysis*

The model fit analysis was performed to show the validity of the suggested model. In this study, the model fit consisted of SRMR, chi-square, and NFI, using model fit parameters from previous studies as a guide [72,73]. As reported in Table 7, all parameter estimates exceeded the minimum threshold value, confirming the proposed model to be valid.

**Table 7.** Model Fit.

| Model Fit for Analysis | Parameter Estimates | Minimum Cut-Off | Recommended by |
|---|---|---|---|
| SRMR | 0.061 | <0.08 | [72] |
| (Adjusted) chi-square/dF | 3.48 | <5.0 | [73] |
| Normal fir index (NFI) | 0.961 | >0.90 | [73] |

*3.4. Result of Final SEM*

The final model for the SEM shown in Figure 3 is based on the initial testing and analysis. The SEM model was evaluated using the beta coefficient and $R^2$ values from the hypothesis test. In total, 21.8% of the variation has resulted in work stress, 67.2% in job satisfaction, and 58.8% was allocated to organizational commitment. Thus, the model was proven adequate to explain or predict the employee's job satisfaction, work stress, and organizational commitment.

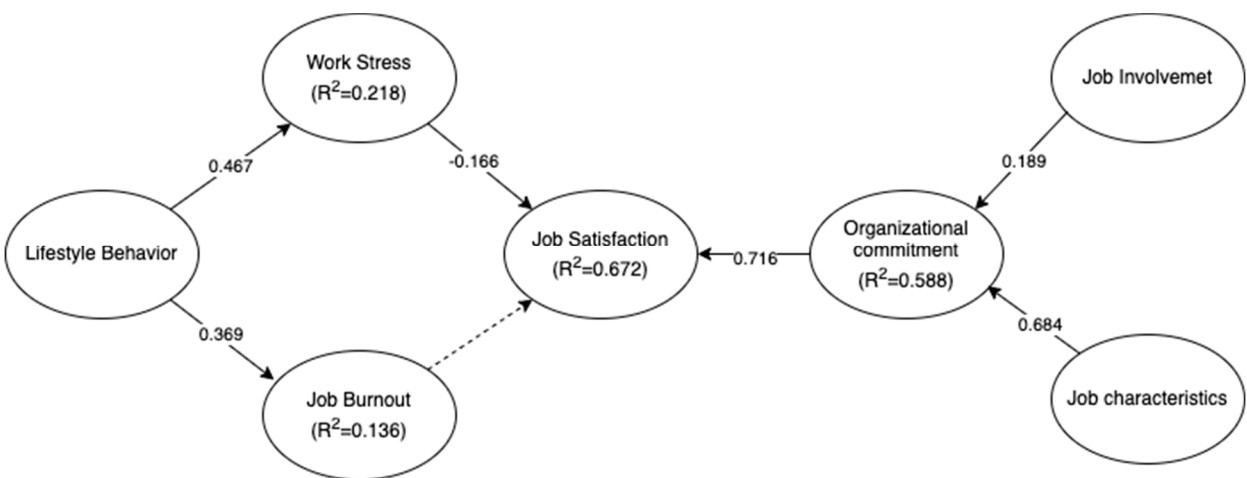

**Figure 3.** Final SEM Model.

**4. Discussion**

By examining the many factors that affect job satisfaction, this study attempted to close a research gap on the subject of supply chain specialists' job satisfaction. Job satisfaction, which has a significant impact on how companies behave, is one of the most crucial components of work motivation. On job satisfaction, there is a wealth of literature and empirical study [30]. Many variables have an impact on job satisfaction, both positively

and adversely. The four most common causes—work stress, job burnout, organizational commitment, and lifestyle choices—have been the subject of this study. Stress at work is a key component of job satisfaction. Stress at work can serve as a catalyst for creativity and fulfillment while erasing routine and boredom. When stress causes work burnout, it breeds resentment and a lack of job satisfaction [32].

In this study, the effect of several factors on workers' job satisfaction were investigated. Some of the aspects that have been researched and employed throughout the model are lifestyle behavior (LB), work stress (WS), job burnout (JB), job involvement (JI), job characteristics (JC), and organizational commitment (OC). The degree of correlation between each element and job satisfaction was calculated using partial least square structural equation modeling (PLS-SEM), proving that each factor is connected to the others.

H1 is supported by evidence that the lifestyle behavior (LB) has a substantial positive association to job stress (= 0.467; *p*-value 0.001). This suggests that unhealthy lifestyle choices have a major impact on work stress. The Collins et al. [81] investigation lends credence to this conclusion. In the same way, this proves that psychosocial stress at work may be a factor that influences or contributes to the adoption or maintenance of a healthy or unhealthy lifestyle. Smoking, heavy alcohol consumption, physical inactivity, and obesity are associated with psychosocial stress at work. Therefore, it is always essential to find time outside of work. It is also essential that a person's lifestyle does not revolve solely around their job but rather that the job is a part of their daily routine. The key is finding hobbies and interests outside of the workplace, such as regular cardiovascular exercise, joining outside organizations, or socializing.

Similarly, job burnout was proved to have a significant positive effect on work stress (β = 0.369; *p*-value = −0.001), thereby supporting H2. Prior studies have also proved the association between job burnout and work stress [82,83]. According to a study, job burnout is a syndrome caused by chronic stress at work, with numerous negative effects on the well-being and health of workers [8]. Shubayr et al. [6] noted that altering one's lifestyle can aid in more effectively managing burnout symptoms. Getting more sleep, practicing mindfulness meditation, taking time off from work, and eating healthy, balanced meals are methods for combating job burnout. In addition, employees must prioritize their health above all else, as it is the most valuable investment they can make.

It was also found that work stress negatively affected job satisfaction (β = −0.166; *p*-value = 0.039), thereby supporting H3. This finding is supported by several studies that demonstrated that work stress affects the job satisfaction and overall performance of employees since most firms today have higher standards for work performance [84–86]. According to Qureshi et al. [87], work stress is a condition in which a person's psychological and physiological state causes them to deviate from their typical behavior. Graham et al. [88] stated in their study that as work stress is associated with job satisfaction, a great indicator of this relationship is workload and age of which younger generations have better work–life balance and work within healthy working hours. The age-claim by Graham et al.'s [88] study supports the fact that 73.5% of the population are aged 31 and up, thus showing an experience of work stress. Further levels of increased work overload can be problematic for individuals and, in the same study, suggests proper staff handling and distribution of load.

The fourth hypothesis, which questions the relationship of job burnout and job satisfaction, happens to result in a Beta Coefficient of −0.139 at a *p* < value of −0.075. The negative sign on the coefficient indicates that the relationship is in a negative state. As job burnout, increases the level of job satisfaction decreases. However, at its *p* < value, the test resulted as not significant, thus rejecting the hypothesis. In a study done on Bangladeshi nurses' job satisfaction, it has shown that job burnout had a significant association with job satisfaction; however, in the same study, it also showed that workplace bullying can associate with job satisfaction similar to job burnout [89]. Among other predictors, job satisfaction may be greatly associated with other known factors not tackled in this study, thus proving this finding to be not significant.

Numerous variables contribute to workplace stress. Hence, an individual needs to recognize the pressures they face in their career. Long-term exposure to environmental and situational stressors that cause work-related stress results in psychological exhaustion, depersonalization, and a lack of personal accomplishment, according to Khamisa et al. [90]. Similarly, work-related stress caused by factors such as increased workloads, staff issues, and a lack of resources have been linked to low job satisfaction. Thus, plans for stress reduction should be developed at individual and group levels. Through education and training, stress management programs that teach individuals how to cope with stress are an individual-level method that assists employees in coping with stress-related impacts. Group-level approaches deal with work-related stressors by lowering or eliminating them through improved personnel management and adequate resources [91]. They have been shown to be effective in organizations where stress is viewed as a natural occurrence that can be managed by fostering an open and understanding culture [92]. To combat work-related stress and job discontent, it is recommended that both an individual and group-centered plan for stress avoidance be implemented. This should involve employee and management input to ensure everyone collaborates to achieve the desired results.

From the results, it could be seen that job characteristics ($\beta$ = 0.684; *p*-value < 0.001) and job involvement ($\beta$ = 0.189; *p*-value = 0.009) have a significant positive effect on organizational commitment, thereby supporting H5 and H6. This explains that job involvement and characteristics are the precursors to the commitment and loyalty of employees to the organization. In the present investigation, job involvement represents the attitude toward work that is typically characterized as the degree of psychological identification with one's work. In contrast, job characteristics are based on the idea that an employee's motivation derives from a specific task. In other words, a challenging, varied job increases motivation, whereas a monotonous, repetitive job causes stress that is detrimental to job satisfaction. Numerous studies support the findings of the present study, which demonstrates a strong correlation between job involvement and characteristics, as well organizational commitment [93].

The study also revealed that organizational commitment ($\beta$ = 0.716; *p*-value < 0.001) significantly influenced workers' job satisfaction, thereby supporting H7. Organizational commitment is a subjective measure that looks at how employees feel they identify with their company's core values, how likely they are to stay with it, and how willing they are to work harder than their company expects. In 2004, Silverthorne [94] stated that job satisfaction significantly predicts organizational commitment. Yiing & Ahmad [47] also found that job satisfaction and organizational commitment are closely linked. Markovits et al. [92] stated that affective organizational commitment was the most crucial factor in both intrinsic and extrinsic levels of job satisfaction. Several organizational characteristics have been linked to organizational commitment, including human resource management techniques, leadership styles, and trust. Since job satisfaction can impact several significant attitudes, intentions, and behaviors, the company should motivate employees to develop attitudes towards aspects of the job such as work, pay, promotion, coworkers, corporate policies, supervisors, and customers. These findings emphasize the importance of components of job satisfaction to organizational commitments. The practical implication of the results is that managers must actively enhance the job satisfaction of their organization's employees to increase organizational commitment.

The entire results collate to the initial model, where each factor associates with one another with indirect relationships, particularly on relationships such as lifestyle behavior on job satisfaction. Although there are studies correlating the two [6], other studies show a stronger and more direct relationship with unhealthy lifestyle behavior to job burnout [43] and work stress [44]. This is also proven within this study of their relationship, with work stress having the weight being affected by unhealthy lifestyle. Within this working model, lifestyle behavior can describe the relationship with job satisfaction; however, results are through work stress but not on job burnout. Since it is found that lifestyle behavior affects work stress significantly, it is at the same time shown that work stress demerits a person's

job satisfaction. With these two findings, there is an indirect link between unhealthy lifestyle behavior and how this can impact a person's job satisfaction.

On the other side of the model, organizational commitment bears the most affecting factor on job satisfaction. Job characteristics contribute more to organizational commitment than job involvement. Thus, providing a healthy work environment, including convenient work schedules, can result in greater organizational commitment and job satisfaction for employees.

## 5. Conclusions

This study examined the elements affecting supply chain employees' job satisfaction using a unique framework for ergonomic appraisal. Purposive sampling was used to create and disseminate a questionnaire to 400 participants. Partially least square structural equation modeling (PLS-SEM) was used to analyze and test the relationships between job satisfaction.

The discriminant validity of the variables has proven that the majority are good and reliable for testing the SEM model. The hypothesis test's beta coefficient and r2 results were used to evaluate the SEM model. The model was found to be adequate to explain or forecast the employee's job satisfaction, work stress, and organizational commitment. It is found that organizational commitment concerning job satisfaction has the value of relationship, among the latent variables work stress accounted for 21.8% of the variation, job satisfaction for 67.2%, and organizational commitment for 58.8%.

Results of the study revealed that organizational commitment ($\beta$ = 0.716; *p*-value < 0.001) had the highest positive association with workers' job satisfaction, while work stress proved to have a negative association with job satisfaction ($\beta$ = $-0.166$; *p*-value = 0.039). Similarly, job characteristics ($\beta$ = 0.684; *p*-value < 0.001) and job involvement ($\beta$ = 0.189; *p*-value = 0.009) were also proved to have a strong positive association with organizational commitment, which in turn influences job satisfaction. Finally, lifestyle behavior was found to have a positive association with work stress ($\beta$ = 0.467; *p*-value < 0.001) and job burnout ($\beta$ = 0.369; *p*-value = 0.001), negatively influencing job satisfaction. Therefore, this study could help the people who work in supply-chain companies learn more about how organizational commitment, lifestyle choices, work stress, and job burnout affect job satisfaction. To make employees satisfied at work, companies need to make sure they are managed well and that their requirements are met.

### 5.1. Practical and Managerial Implications

Businesses worldwide are making every effort to recover from the current COVID-19 outbreak. The quick transformation resulted in systems undergoing abrupt modifications and frequently adapting new regulations. As a result, this study can offer supply-chain company personnel more information on the effect of organizational commitment, lifestyle behavior, work stress, and job burnout on job satisfaction. Companies must ensure that employees are treated well and meet their demands to further improve their job satisfaction after learning that organizational commitment has a strong relationship to job satisfaction. Their job characteristics contribute the most to affecting an employee's organizational commitment.

The level of job satisfaction can be improved based on the identified attributes and calculated for any organization, or a standard can be used to enhance the quality of the working environment in order to make employees satisfied. Job satisfaction is contingent on several variables and can only be determined at the ground level by the human resource department of a specific industry. Thus, it is important for supply chain and logistic employers to understand how the identified factors affect employee satisfaction. This study may have provided relevant information for other industries with comparable characteristics to consider. Shown on Table 8 are the recommended programs for the employers.

**Table 8.** Recommendations.

| To Whom | Target | Recommended Programs |
| --- | --- | --- |
| Employees | -Avoid unhealthy habits<br>-Exercise more | Attend aerobic activities, consult dietitians for healthy consumptions, get proper sleep |
| Employers | -Workload Distribution<br>-Employee–work communication | -Provide occasional retreats for the team, one on one talk with employees for individual assessments regarding job involvement.<br>-Create safe workplace for employees, routine inspection of office equipment and environment. |

*5.2. Limitations and Future Use*

This study was limited only to supply-chain management employees in the Philippines and found that the most effective value of the population was 300 or more respondents. Despite this study's positive outcomes, the respondents who participated may not represent the population completely as the working survey was handed out and filled out by participants who allowed the data gathering on their premises. More in-depth data may be concluded once the researcher has permission to perform data gathering to more organizations. It is also worth taking note of the time in performing the data gathering, as due to the season having multiple holidays, workers tend to have deviating schedules and heavy workloads which may affect the accuracy of the data.

Future research can address age, gender, residence, education, and salary as moderating effect. This future research directions could help expand our understanding of the moderating effects of demographic variables. These directions can address the limitations and provide more comprehensive insights into the relationships within the framework. Investigating potential mediating variables can help uncover the mechanisms through which demographic variables influence the relationships within the framework. Identifying and examining these mediating variables can offer deeper insights into the underlying processes and pathways involved. Future researchers may use the newfound conclusions and data or adapt this study with further analysis and considerations.

**Author Contributions:** Conceptualization, M.J.J.G. and C.K.K.I.; Software, C.K.K.I.; Validation, M.J.J.G.; Formal analysis, M.J.J.G.; Investigation, M.J.J.G.; Resources, C.K.K.I.; Data curation, C.K.K.I.; Writing—original draft, C.K.K.I.; Writing—review & editing, M.J.J.G.; Supervision, M.J.J.G.; Funding acquisition, M.J.J.G. All authors have read and agreed to the published version of the manuscript.

**Funding:** This research was funded by Mapúa University Directed Research for Innovation and Value Enhancement (DRIVE) (Funding No. FM-RC-21-89).

**Institutional Review Board Statement:** The study was approved by Mapúa University Research Ethics Committees (FM-RC-21-89).

**Informed Consent Statement:** Informed consent was obtained from all subjects involved in the study (FM-RC-21-89).

**Data Availability Statement:** The data presented in this study are available on request from the corresponding author.

**Conflicts of Interest:** The authors declare no conflict of interest.

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
