# Peer review of "The Impact of Job Satisfaction on Creating a Sustainable Workplace: An Empirical Analysis of Organizational Commitment and Lifestyle Behavior"

_sustainability, doi:10.3390/su151310283_

Round 1

Reviewer 1 Report

I found the title and the selection of variables in the conceptual framework were not sufficiently described to indicate the focus of the main issue in logistic. Does Job Satisfaction a dependent variable in this case or organizational commitment? Or do you have two dependent variables?

Are you testing JS as mediator; or work stress and job burnout are mediators between lifestyle behavior and JS? Include any relevant theory to support your conceptual framework for logistics employees. It seems that all your hypotheses are direct relationships and so does your findings in Table 6. What is the point of using PLS-SEM if you don't examine path analysis for testing mediation like you have mentioned in lines 240-244?

I don't see any major difference between before and after covid since your constructs and measurements have been adopted from previous studies that are commonly appeared and studied. When you mentioned about flexible work arrangements during covid (line 44-45 etc), how this change at work could be measured for logistics?

How could you describe your ethical considerations in data collection procedures to overcome common method variance issue?  Explain sampling procedures and response rate. What is the total number of your usable responses that you used in your data analysis? 400 out of of 400?

Reviewer 2 Report

Dear autors:

I would like to provide some recommendations for improving the quality and clarity of the research paper:

I kindly request that you include concise definitions of key concepts in the introduction. Specifically, please provide definitions for important terms such as burnout, work stress, organizational commitment, job characteristics, job involvement, and lifestyle behavior. By ensuring that these concepts are clearly defined, readers will have a better understanding of the relevance and significance of your study.

I would like to highlight the importance of analyzing whether the demographic information presented in Section 3.1 Demographic Profile, such as gender, age, educational level, residential area, job position, employment status, and monthly salary, may have a significant effect on the obtained results in your study. Specifically, I would like to emphasize the potential influence of generational factors, such as Millennials, Generation X, etc., which could introduce bias in the results. I recommend conducting additional analysis or discussing how these demographic factors might influence the study results. Exploring the potential impact of generation on participants' attitudes, behaviors, or perspectives could provide a deeper understanding and a more comprehensive interpretation of the findings.

I would like to bring to your attention a discrepancy in the presentation of age groups and their corresponding percentages in lines 250-251. According to Table 2, the correct order of age groups and their percentages should be as follows: the majority of respondents, representing 38%, fall within the 31-40 age range, followed by 28% for the 41-50 age group, and 26% for the 21-30 age group. However, the current order in the text does not match the information in Table 2.

I recommend that the authors update the references used in the study and be cautious about relying heavily on classical references, as some of the cited sources date back to 1981. It is important to ensure that the references are current and up-to-date to adequately support the arguments and findings presented. Additionally, diversifying the sources and incorporating more recent research will strengthen the analysis and conclusions of the study. Please consider revising the references to include more recent sources and avoiding an over-reliance on older references. This will enhance the validity and relevance of the study.

Regards.

Reviewer 3 Report

This is a very strong article of special relevance with so many dramatic challenges in workplaces coping with Covid-19 and its aftermath.

The literature review is very strong, utilizing 85 references! The literature review is clear and is especially strong in defining job burnout,  job satisfaction, and job dissatisfaction. The Conceptual Framework is a significant addition and orients the readers toward the 4 most common issues related to job satisfaction: work stress, job burnout, organization commitment, and lifestyle behavior. Figure 1 nicely illustrates the conceptual framework. Section 2.2 Determinants of Job Satisfaction is well-done and the reader can easily see and understand the 7 hypotheses.

400 respondents is a very strong sample. This section needs a bit of fine-tuning to reflect that it is one questionnaire in two parts. It needs to be clarified if the authors devised the 44-item questionnaire.

Line # 208: should be "44-item questionnaire"...

Table 1 gives a good overview of the measurements items. It is a bonus to also include supporting references in the table.

The demographic profile reflects a relatively good balance between men and women (54% vs. 46%). We learn ages, education level, geographical areas and work positions, including salaries, of respondents. Table 2 nicely categorizes the demographics.

I am not a research methodologist/statistician and hope that another reviewer can be helpful here. I get that the final model for the SEM is adequate to explain or predict job satisfaction, work stress, and organizational commitment. While challenged by the statistical data, I was able to glean the important findings from simpler Discussion section.

The first 3 hypotheses are supported and the 4th hypothesis, addressing job burnout and job satisfaction, is not supported. The authors nicely point out that other unaddressed factors can have an influence. Hypotheses 5, 6, and 7 are clearly supported and this discussion is very clear.

Line # 466: "Which" should be "This"...

Line # 474: "contributes" should be "contribute"

Line #s 474-477 is better as: "Thus, providing a healthy work environment, including convenient work schedules, can result in greater organizational commitment and job satisfaction for employees.

I especially appreciate Section 5.1 Practical and Managerial Implications. These specific recommendations are very helpful. The end of the article offered good recommendations for other researchers.

Overall, a very strong effort!

Line #s 66-70 are better as: "Job burnout is characterized by feelings of emotional exhaustion, cynicism, and a lack of effectiveness at work. According to Maslach et al. [9], burnout been linked to several adverse affective outcomes, such as a decline in job satisfaction, which can negatively impact work performance, loyalty to the organization, and employees’ desire to remain in the organization.

Line #122: insert "the" before "main objective"...

Turn the limitations section into a positive.

The last paragraph is better in the positive. "Future research can address age, gender, education and salary."

English quality is good. I couldn't resist making a few very few minor corrections.

Round 2

Reviewer 1 Report

I appreciate your explanation to answer my previous comments and would be grateful if the justification in question 3 and 4 could be added in the text to clarify your idea and writing for logistics' nature of work during pandemic and the validation of your sample selection. What is your response rate? It will be useful if you may also add any underlying theory involved as your data analysis tested direct relationship between several IVs and DV. 
